

# 'Degraded' RNA profiles in Arthropoda and beyond

Sean D. McCarthy, Michel M. Dugon and Anne Marie Power

School of Natural Sciences, Ryan Institute for Environmental, Marine and Energy Research, National University of Ireland Galway, Ireland

## ABSTRACT

The requirement for high quality/non-degraded RNA is essential for an array of molecular biology analyses. When analysing the integrity of rRNA from the barnacle *Lepas anatifera* (Phylum Arthropoda, Subphylum Crustacea), atypical or sub-optimal rRNA profiles that were apparently degraded were observed on a bioanalyser electropherogram. It was subsequently discovered that the rRNA was not degraded, but arose due to a 'gap deletion' (also referred to as 'hidden break') in the 28S rRNA. An apparent excision at this site caused the 28S rRNA to fragment under heat-denaturing conditions and migrate along with the 18S rRNA, superficially presenting a 'degraded' appearance. Examination of the literature showed similar observations in a small number of older studies in insects; however, reading across multiple disciplines suggests that this is a wider issue that occurs across the Animalia and beyond. The current study shows that the 28S rRNA anomaly goes far beyond insects within the Arthropoda and is widespread within this phylum. We confirm that the anomaly is associated with thermal conversion because gap-deletion patterns were observed in heat-denatured samples but not in gels with formaldehyde-denaturing.

## INTRODUCTION

Anomalies in the gel migration of the 28S subunit rRNA in denatured samples have mostly appeared in the older literature (*Applebaum, Ebstein & Wyatt, 1966*; *Ishikawa & Newburgh, 1972*; *Fujiwara & Ishikawa, 1986*). More recent studies have also mentioned anomalies observed during routine RNA extraction and quality control steps using both automated (*Winnebeck, Millar & Warman, 2010*; *Asai et al., 2014*) and traditional (*Macharia, Ombura & Aroko, 2015*) electrophoresis methods. The Agilent bioanalyser is the gold standard for RNA quality analysis prior to deep sequencing and gene expression experiments. This method generates an electropherogram showing an RNA profile with peaks representing the various subunit components, as well as an RNA integrity number (RIN). This analysis is routinely carried out following RNA denaturation using a heating step (*Krupp, 2005*). An alternative to the bioanalyser is using an electrophoresis denaturing gel to confirm the integrity of the RNA sample. This also requires a heating step to prevent secondary structure formation. Several denaturing reagents are commonly used for this including methyl mercury hydroxide, urea, guanidine thiocyanate, formamide, DMSO and

Corresponding author
Anne Marie Power,
Annemarie.power@nuigalway.ie

formaldehyde, with the latter being used most frequently (*Masek et al., 2005*). The problem that arises is that, for some groups of organisms, accurate analysis of the RNA quality is impossible unless non-heat-denaturing steps are followed.

In most animals, RNA denaturation generates 28S and 18S rRNA fragments (~4,000 and ~2,000 nucleotides (nt) respectively), which migrate separately on the denaturing gel. It is widely accepted that, in intact rRNA, the 28S and 18S peak ratios must be close to 2:1 (*Skrypina et al., 2003*). But this is not the case for all groups: in the 1970s, studies on the 28S rRNA in insects found that heat-denaturing RNA for several minutes resulted in just a single band, explained by a splitting of the 28S rRNA into two fragments '$\alpha$' and '$\beta$' (*Ishikawa & Newburgh, 1972*; *Ishikawa, 1976*) with this split likely occurring in the variable D7a region cleavage site (*Gillespie et al., 2006*). At the cleavage site, heat-denaturing the hydrogen bonds holding $\alpha$ and $\beta$ fragments together, causes these to migrate together with the 18S fragment because all three fragments (18S, $\alpha$, and $\beta$) are similar in size and form a single band on RNA gels, approximately at the 18S position. State-of-the-art RNA profile analysis using bioanalyser electropherograms can also display a 'degraded' appearance in RNA due to the presence of a single peak ~ at the 18S position, as previously described by *Winnebeck, Millar & Warman (2010)*.

Very often, the anomalous RNA profile described above has been discovered by accident during quality control steps and this phenomenon has not been subjected to detailed study, with some notable exceptions (*Fujiwara & Ishikawa, 1986*; *Ogino et al., 1990*; *Sun et al., 2012*). This phenomenon has mostly been described as a 'hidden break' (*Ishikawa & Newburgh, 1972*; *Ishikawa, 1977*; *Fujiwara & Ishikawa, 1986*; *Ogino et al., 1990*; *Winnebeck, Millar & Warman, 2010*; *Macharia, Ombura & Aroko, 2015*), but may more correctly be termed a 'gap deletion' because the cleavage is actually due to the excision of a short sequence from the rRNA precursors (*Ware et al., 1983*; *Sun et al., 2012*). Several authors have wondered why this observation is limited to insects (*Ogino et al., 1990*; *Winnebeck, Millar & Warman, 2010*) and have recommended further study (*Sun et al., 2012*). Others have suggested that the gap deletion occurs more widely—e.g., in most protostomes (*Fujiwara & Ishikawa, 1986*); however, there are exceptions of individual species within a phylum lacking the deletion: e.g., *Caenorhabditis elegans* in Phylum Nematoda (*Zarlenga & Dame, 1992*) and aphids in class Insecta (*Ogino et al., 1990*).

Clearly, this question has both practical and theoretical relevance since more knowledge will save researchers a lot of time during optimisation and quality control of RNA extractions in a variety of animal groups. It could also provide important information about the maturation of rRNAs, molecules that are fundamentally important in the field of evolution and systematics, as well as being associated with certain genetic diseases (*Henras et al., 2015*). The present study came about during routine RNA extractions steps in stalked barnacles whilst investigating bioadhesive genes (*Jonker et al., 2012*; *Jonker et al., 2014*). This study is the first documentation of a gap deletion in non-insect arthropods including stalked barnacles (Crustacea), spiders (Chelicerata) and centipedes (Myriapoda). In addition, we review the prevalence of this gap deletion in groups of animals beyond the arthropods, as well as in non-animal organisms.

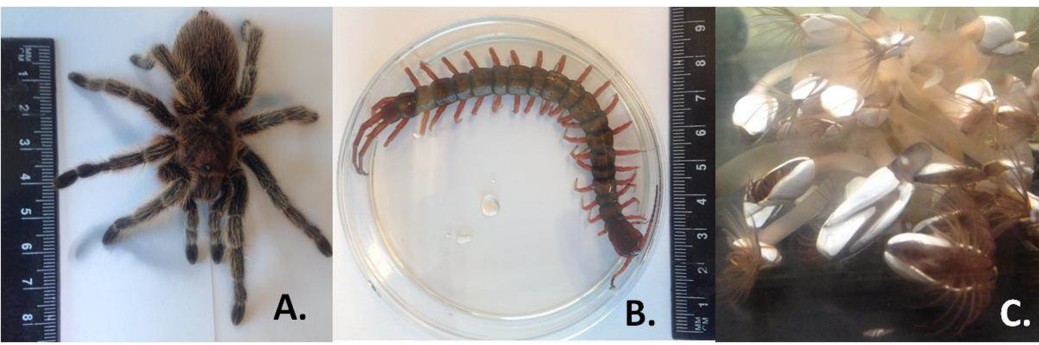

**Figure 1 Species analysed for the gap deletion.** (A) Spider *Grammostola porteri*; (B) Centipede *Scolopendra subspinipes*; and C) Barnacle *Lepas anatifera*.

## MATERIALS AND METHODS

Tissue samples were obtained from Chilean rose tarantula *Grammostola porteri* (Chelicerata) and giant Asian centipede *Scolopendra subspinipes* (Myriapoda). Both of these organisms are not easily obtained and so we were limited to one individual specimen for each ($n = 1$); however, RNA was extracted from three replicate tissue samples from each individual. For barnacles (Crustacea), three species in different taxonomic orders were sampled, this time with tissues from three individuals of each species ($n = 3$). Specimens were instantly killed and frozen by immersion in liquid nitrogen. Immediately afterwards, specimens were left to thaw for a few minutes before being dissected. Approximately 2 cm$^3$ of tissue was excised from the muscle mass surrounding the fovea on the lateral parts of the cephalothorax of *Grammostola porteri* and divided into three samples for analysis. Approximately 1 cm$^3$ of tissue was recovered from the deep oblique and lateral longitudinal muscles of *Scolopendra subspinipes* and then divided into three samples. Approximately 1 cm$^3$ of prosoma and peduncle tissues were dissected from each of three individuals of *Lepas anatifera* (Fig. 1), *Dosima fascicularis* and *Pollicipes pollicipes* barnacles. Tissues from all specimens were snap-frozen in liquid nitrogen, and stored at −80 °C.

Tissues were homogenized using the IKA® ULTRA-TURRAX T18 homogeniser in 1 ml RLT lysis buffer with 10 ul beta-mercaptoethanol (QIAGEN RNeasy Mini Kit; Qiagen, Hilden Germany). Next, total RNA was extracted with QIAGEN RNeasy Mini Kit including DNase I treatment (QIAGEN DNase I, RNase-free), according to the manufacturer's instructions. For the analyses of quality and integrity, RNA (100–200 ng) was electrophoretically separated with an Agilent 2100 Bioanalyser using an RNA 6000 Nano Chip Kit according to manufacturer's instructions. Each sample was heat-denatured at 70 °C for 3 min prior to separation. Aliquots from the same RNA samples without heat-denaturation were also run together with heat-denatured samples on the same Bioanalyser chip.

Samples were also run on a formaldehyde-denaturing RNA gel. One gram of agarose was added to 90 ml of nuclease free water and transferred to a 50 °C water bath until equilibrated. In a fume hood, 10 ml of NorthernMax® 10X Gel Buffer (Thermo Fisher

Scientific, Inc., Waltham, Massachusetts, USA) was added to the agarose gel solution and mixed. The gel was poured to a thickness of 0.6 cm and allowed to set. The gel was run at 60 V using 1X NorthernMax® MOPS Running Buffer (Thermo Fisher Scientific, Inc., Waltham, Massachusetts, USA) for 90 min. Aliquots from the same RNA stock were run in both heat-denatured (70 °C for 3 min) and non-heated form, along with RiboRuler High Range RNA Ladder (Thermo Fisher Scientific, Inc., Waltham, Massachusetts, USA). The procedure was also carried out on a stronger denaturing gel (2X), i.e., 20 ml of NorthernMax® 10X Gel Buffer was added to 1 g of agarose and 80 ml of nuclease free water and this was run as before with the respective RNA samples.

Deep sequencing by RNA-seq of *Lepas anatifera* RNA was carried out as part of a parallel study (SD McCarthy et al., 2015, unpublished data). Transcriptome analysis for this species was carried out on an Illumina Hi-Seq 2000 sequencing platform by Source BioScience LifeSciences, UK. Sequence alignments of 28S rDNA gene sequences were carried out using the *Lepas anatifera* 28S gene sequence from the transcriptome study, along with all available sequences from GenBank database. The GenBank accession numbers for sequences were: *Scolopendra subspinipes* (HQ402538.1), *Dosima fascicularis* (KF781345.1) and *Pollicipes pollicipes* (EU370441.1). The online tools Clustal Omega (http://www.ebi.ac.uk/Tools/msa/clustalo/) and T-COFFEE Multiple Sequence Alignment Server (http://tcoffee.crg.cat/apps/tcoffee/index.html) were used to align sequences and search for common motifs.

## RESULTS AND DISCUSSION

The results showed evidence of a gap deletion in 28S rRNA of all five of the Arthropod species investigated. Bioanalyser electropherograms clearly showed no 28S rRNA peak (i.e., a 'degraded' profile) in heat-denatured samples in all cases (Fig. 2). These species were deliberately chosen to represent the different clades of arthropod 'classes' (*Rota-Stabelli et al., 2011*) i.e., Crustacea (barnacle), Chelicerata (spider) and Myriapoda (centipede). A gap deletion was previously described in the Hexapoda (insects) (*Fujiwara & Ishikawa, 1986*; *Ishikawa & Newburgh, 1972*), which is also within the Phylum Arthropoda. Thus, clear evidence of the deletion exists throughout the Phylum Arthropoda. The fact that the pattern was seen in all three species of barnacle in different taxonomic orders—i.e., *Lepas anatifera* and *Dosima fascicularis* (Order Lepadiformes) as well as *Pollicipes pollicipes* (Order Scalpelliformes—*Jonker et al., 2014*)—indicates that the gap deletion may be prevalent throughout this group rather than being restricted to individual species.

In all groups that displayed the rRNA gap deletion, heat-denaturation affected the 28S rRNA so that this resembled the 18S rRNA in size and therefore appeared with 18S as a single peak on the bioanalyser, or as a single band on a gel (Fig. 3). Superficially, this appeared like degraded RNA; however, this was not the case because bioanalyser-generated RNA Integrity Numbers (RINs) were high (>8.0) for aliquots from the same sample stock when analysed in a non-heat-denatured state (Fig. 2). The effect of a denaturing gel alone (without heating) did not affect the RNA in the same way as samples run on denaturing gels did not show gap deletions (Fig. 3; lanes 2–6). This remained the case even

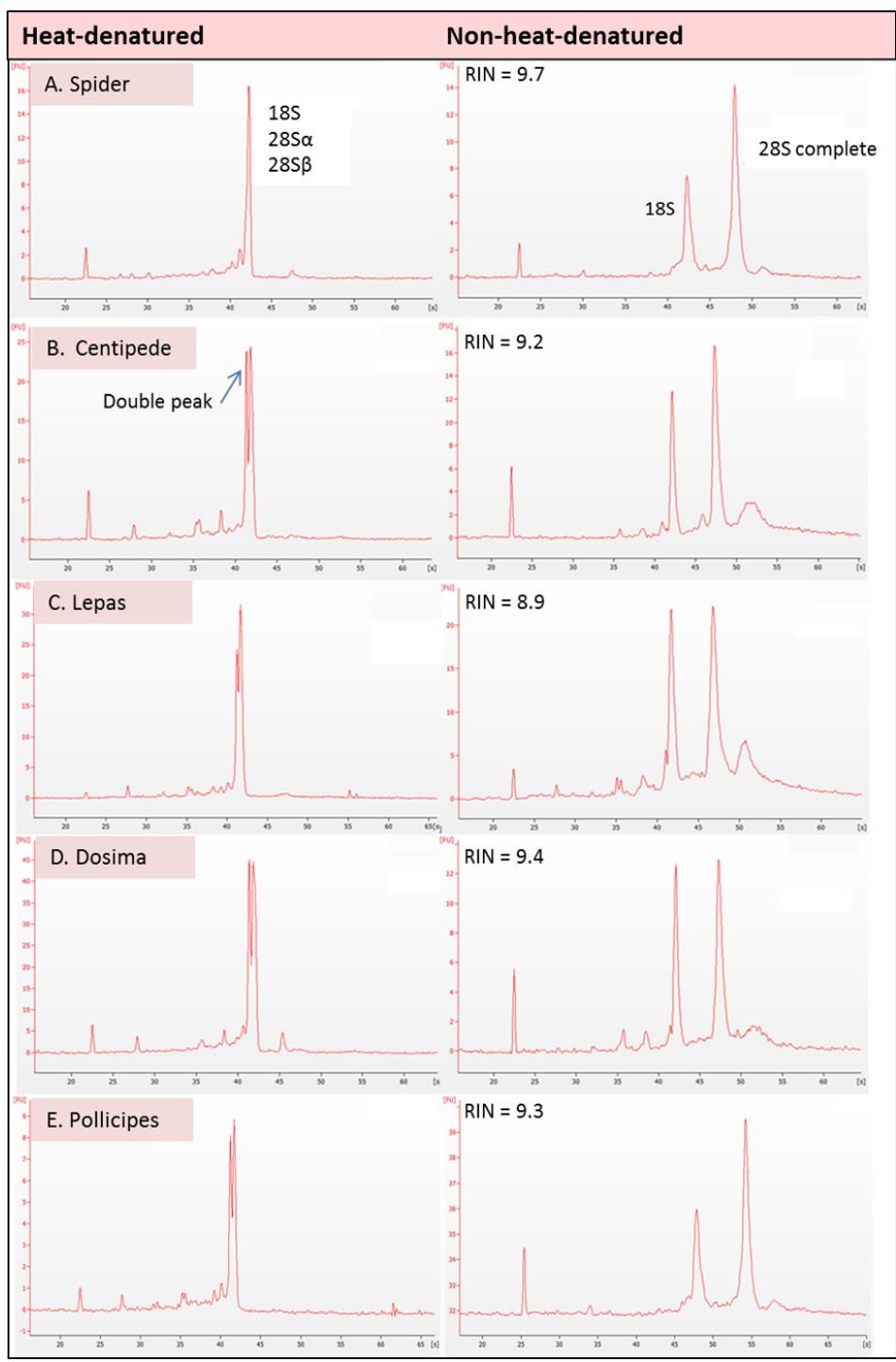

**Figure 2** **Electropherogram traces for 100–200 ng of total RNA applied to an RNA Nano Chip were generated on the Agilent 2100 Bioanalyser.** RNA that appears 'degraded' after heat-denaturation and fails to provide a RNA Integrity Number (RIN) can generate high RINs in non-heat-denatured aliquots from the same RNA stock. RIN numbers are shown in each case for (A) Spider *Grammostola porteri*; (B) Centipede *Scolopendra subspinipes*; (C) Stalked barnacle, Order Lepadiformes, *Lepas anatifera*; D. Stalked barnacle, Order Lepadiformes, *Dosima fascicularis* E. Stalked barnacle, Order Scalpelliformes, *Pollicipes pollicipes*.

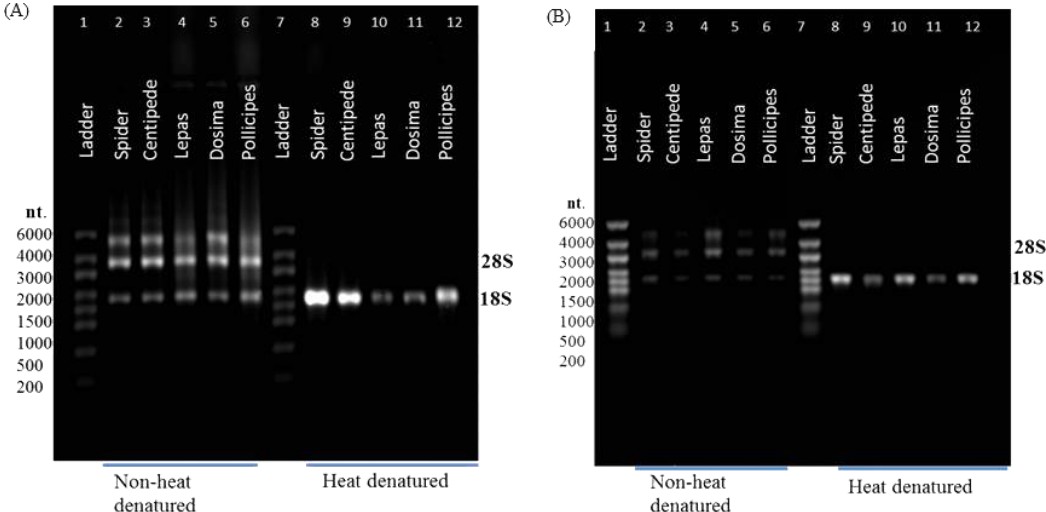

**Figure 3 rRNA migration patterns on denaturing gel.** Non-heat-denatured samples (lanes 2–6) and denaturing gel with heat-denatured samples (3 mins at 70 °C) (lanes 8–12) using (A) 1X NorthernMax® denaturing Gel Buffer and (B) 2X NorthernMax® denaturing Gel Buffer. Extra bands in non-heat-denatured samples are likely to be due to secondary structures. RNA samples are from spider (lanes 2 & 8), centipede (lanes 3 & 9) and stalked barnacles (lanes 4–6 & lanes 10–12).

when denaturant concentration was doubled. Therefore, the gap deletion profile certainly arises due to thermal conversion of the 28S rRNA. Bioanalyser protocols recommend heat-denaturation of RNA prior to analysis (*Krupp, 2005*), but based on our findings, heating the RNA samples may best be avoided for correct interpretation of RNA quality in certain groups. In non-heat-denatured samples, a band appeared just below 6,000 nt following electrophoresis. This band did not appear on heat-denatured samples from the same sample stock, indicating that this artefact is almost certainly secondary structure formation, which is prevented by the heating step (Fig. 3).

The Agilent 2100 Bioanalyser is an established method for determining RNA quality and integrity for the purpose of downstream transcriptomic analysis. But several authors have remarked on a degree of puzzlement in interpreting bioanalyser electropherograms where gap deletions are involved (*Winnebeck, Millar & Warman, 2010*) and new groups containing gap deletions are appearing all the time (e.g., *Asai et al., 2014*). This demonstrates how gap deletion profiles in rRNA (i) are relatively unknown in the literature (ii) have unknown prevalence across groups of different organisms and (iii) can present as a problem during routine RNA extraction and quality control protocols and can waste a lot of time. Thus, we undertook a literature review to find how widespread the gap deletion pattern actually is, especially since there are exceptions within some groups (see Table 1).

The literature to date has either suggested that the gap deletion (or hidden break) is very narrowly focussed e.g., within 'insects' (*Winnebeck, Millar & Warman, 2010*), or that it is nearly ubiquitous. According to a review by *Ishikawa (1977)*, almost all the protostome animals possess the hidden break, whereas all the deuterostome animals do not (see Table 2 in *Ishikawa, 1977*). Table 1 updates the information provided by *Ishikawa (1977)* drawing on information across the biological domains. Non-protostome animals including sponges

**Table 1** Prevalence of gap deletion in the LSU rRNA molecule across taxa including Bacteria and Eukaryota domains.

| | Gap | LSU rRNA | Phylum | Clade | Reference |
|---|---|---|---|---|---|
| Higher plants (angiosperms) | Yes | 25S | n/a | n/a | *Fujiwara & Ishikawa (1986)* |
| | No | | | | *Krupp (2005)* |
| *Actinobacillus Actimomycetemcomitans* | Yes | 23S | Proteobacteria | Gamma-proteobacteria | *Haraszthy et al. (1992)* |
| [*] *Escherichia coli* | Yes | 23S | | | *Ishikawa & Newburgh (1972)* |
| [*] *Escherichia coli* | No | 23S | Proteobacteria | Gamma-proteobacteria | *Krupp (2005)* |
| *Tetrahymena pyriformis* | Yes | 26S | Ciliophora | Oligo-hymenophorea | *Eckert et al. (1978)* |
| *Acanthamoeba castellanii* | Yes | 26S | Protozoa | Amoebozoa | *Stevens & Pachler (1972)* |
| *Haliclona indistincta* | No | 28S | Porifera | Demospongiae | This study (Fig. S2) |
| *Halichondria japonica* | No | 28S | Porifera | Demospongiae | *Ishikawa (1975a)* |
| [*] *Actinia equina* | Yes/No | 28S | Cnidaria | Anthozoa | *Ishikawa (1975b)* |
| *Dugesia japonica* | Yes | 28S | Platyhelminthes | Tricladida | *Sun et al. (2012)* |
| *Schistosoma mansoni* | Yes | 28S | Platyhelminthes | Trematoda | *Van Keulen et al. (1991)* |
| *Caenorhabditis elegans* | No | 28S | Nematoda | Chromadorea | *Zarlenga & Dame (1992)*; *Krupp (2005)* |
| *Trichinella spiralis* | Yes | 28S | Nematoda | Adenophorea | *Zarlenga & Dame (1992)* |
| *Grammostola porteri* | Yes | 28S | Arthropoda | Chelicerata | This study |
| *Artemia parthenogenetica* | Yes | 28S | Arthropoda | Crustacea | *Sun et al. (2012)* |
| *Dosima fascicularis* | Yes | 28S | Arthropoda | Crustacea | This study |
| *Lepas anatifera* | Yes | 28S | Arthropoda | Crustacea | This study |
| *Pollicipes pollicipes* | Yes | 28S | Arthropoda | Crustacea | This study |
| *Apis mellifera* | Yes | 28S | Arthropoda | Hexapoda | *Winnebeck, Millar & Warman (2010)* |
| *Bombyx mori* | Yes | 28S | Arthropoda | Hexapoda | *Fujiwara & Ishikawa (1986)* |
| *Thrips tabacai* | No | 28S | Arthropoda | Hexapoda | *Macharia, Ombura & Aroko (2015)* |
| *Scolopendra subspinipes* | Yes | 28S | Arthropoda | Myriapoda | This study |
| *Xenopus laevis* | No | 28S | Chordata | Amphibia | *Ware et al. (1983)* |
| Avian/mammal | No | 28S | Chordata | Aves/Mammalia | *Krupp (2005)* |
| [**] *Ctenomys sp.* | Yes | 28S | Chordata | Mammalia | *Melen et al. (1999)* |

**Notes.**

[*] Presence of denaturing induced gap deletion not well established (conflicting evidence).

[**] Exception in the Deuterostomia.

(Phylum Porifera) and some members of the Phylum Cnidaria appear to lack the gap deletion; similarly, unpublished observations in our laboratory showed no gap deletion in sponge rRNA (*Haliclona indistincta*) (Fig. S2). Meanwhile, protostomia such as the platyhelminths *Schistosoma mansoni* (*Van Keulen et al., 1991*) and *Dugesia japonica* (*Sun et al., 2012*) contained the gap deletion, as did the Arthropoda. There is apparently one case of a gap deletion in mammalian/deuterostome RNA (*Melen et al., 1999*; Table 1), but since this deletion is not in the same position on the molecule as in other groups, it does not lead to the 'degraded' appearance which is the focus of the present study. Still the suggestion of a less obvious deletion hints that deletions in deuterostomes may be more prevalent than

current studies suggest. The gap deletion is therefore almost ubiquitous in protostome groups and its prevalence is scattered across other animal clades. It should be noted that there are also some exceptional groups of protostomes lacking the gap deletion; e.g., within the insects, the aphids lack the deletion (*Ishikawa, 1977*) as do certain Nematoda (Table 1).

Further afield, within Eukaryota the gap appears to be found in some single celled organisms and even higher plants (*Applebaum, Ebstein & Wyatt, 1966*; *Eckert et al., 1978*; *Fujiwara & Ishikawa, 1986*). For example, gap deletions may be found in ciliophore (*Tetrahymena* spp.) and amoeboid LSU rRNA (*Eckert et al., 1978*; *Ishikawa & Newburgh, 1972*). It may even occur in LSU rRNAs across several biological domains. Within the bacteria, the proteobacterium *Actinobacillus actinomycetemcomitans* may also display a gap deletion (*Haraszthy et al., 1992*), though evidence in *Escherichia coli* is more ambiguous (Table 1).

*Gould (1967)* and *Ishikawa & Newburgh (1972)* were the first to suggest thermo-conversion of rRNA in insects and our results support this observation, as gap deletions were seen in heat-denatured arthropod samples, whereas non-heat-denaturing methods alone (denaturing gels) did not have this effect (Fig. 3). Experiments involving urea-denaturation may have used heat simultaneously and these are difficult to interpret in this context (*Ishikawa & Newburgh, 1972*; *Summer, Gramer & Droge, 2009*; *Sun et al., 2012*). However, our results suggest that formaldehyde denaturation did not have a gap deletion effect, even at higher concentration (Fig. 3B). A normal RNA migration pattern will only be revealed in these circumstances by skipping the heat-denaturation step in both Bioanalyser and electrophoresis gel analyses (Fig. 4). Doing this allows a normal migration profile to be observed and concerns about RNA degradation may be allayed, although the appearance of large fragments associated with secondary RNA structures may present on gels in these instances. Figure 4 outlines our recommended procedure when dealing with RNA in taxa containing gap deletions.

The mechanism for the 'hidden break' (=gap deletion) was proposed initially by *Ishikawa & Newburgh (1972)* as disruption of the covalent phospho-ester bonds conferring secondary structure to the RNA during denaturation, leading to dissociation of the large rRNA molecule into two smaller sub-units. These authors also proposed the above came about due to the melting of hydrogen bonds, as H-bond strength is proportional to the strength of chemical changes under denaturing conditions. A later study by *Fujiwara & Ishikawa (1986)* suggested that it is the AU-rich D7a expansion segment of the 28S molecule that is the site responsible for the gap deletion. In the bee *Apis mellifera*, this cleavage site is located in the D7a hairpin loop around position 1,350 nt. The majority of the D7a heterogeneity is located at the distal region of helix D7a-3, where the sequences are highly variable in length and base composition (*Gillespie, Yoder & Wharton, 2005*). Here the double bonded AU pairing is weaker than the triple bonded GC association through their H-bonds and also through base-stacking interactions (*Mallatt & Chittenden, 2014*; *Yakovchuk, Protozanova & Frank-Kamenetskii, 2006*). Thus, GC-rich sequence tracts in the D7a expansion segment are associated with the groups lacking the gap processing (*Fujiwara & Ishikawa, 1986*). Recently, *Sun et al. (2012)* examined the predicted secondary structure of the pre-processed molecule and also the primary (AU-rich) sequence at the

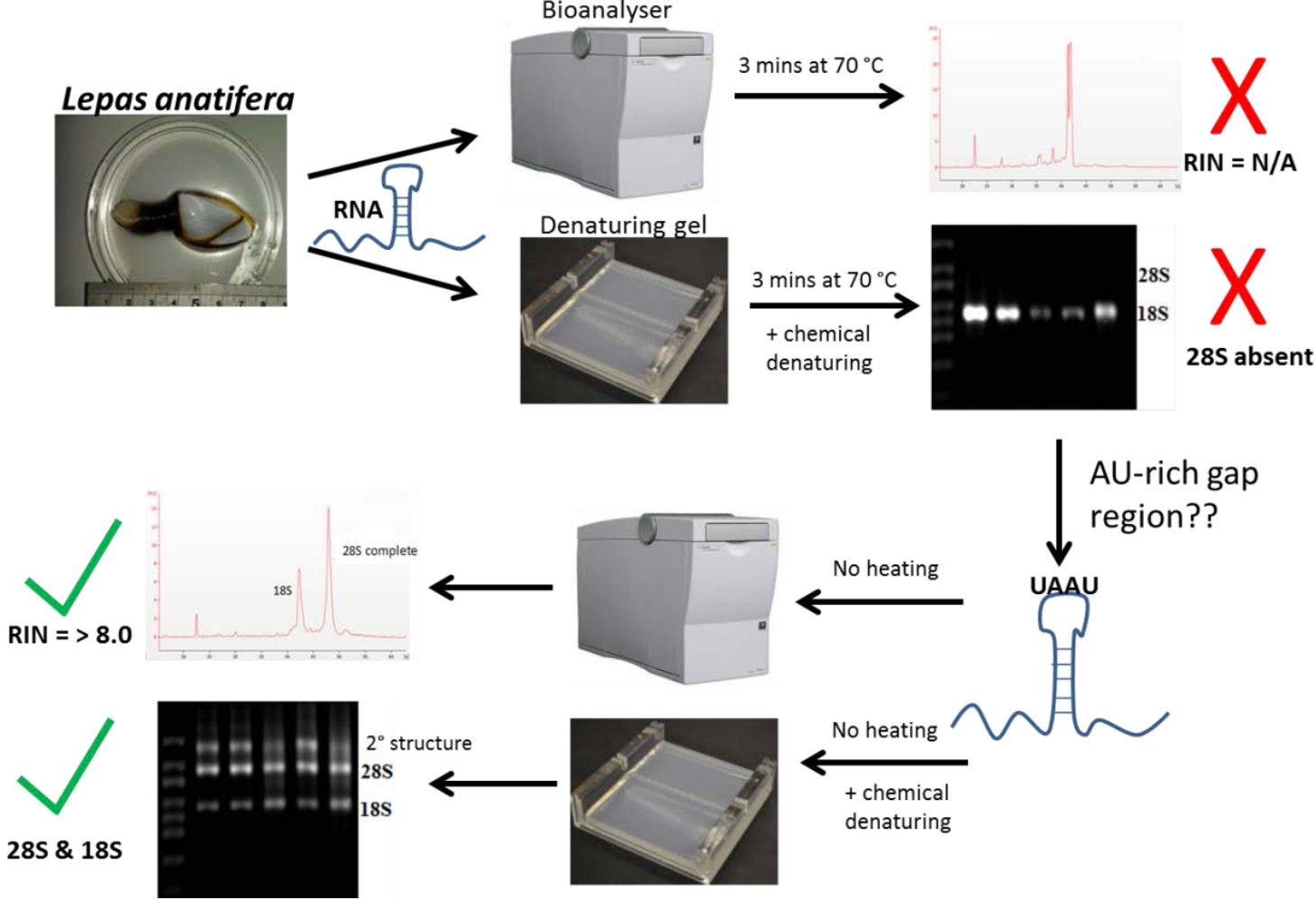

**Figure 4 The steps involved in determining RNA quality in a species containing a 'gap deletion'.** These include Arthropoda, most protostome animals and a scattering of other groups (-see Table 1). The schematic displays the problem with taking routine approaches in taxa with gap deletions because standard protocols specify heat-denaturation of the rRNA prior to Bioanalyser analysis. The latter measures migration and intensity of LSUs and RNA Integrity Numbers (RINs). Heat-denaturation prevents visualisation of the 28S peak in Bioanalyser electropherograms in the affected taxa, so that RNA appears 'degraded' and a RIN cannot be generated. The solution is to run non-heat-denatured rRNA during Bioanalyser analysis. Denaturing formaldehyde gel electrophoresis of rRNA does not appear to bring about the gap deletion and RNA subunits appear normally on these gels.

gap site. During the maturation of the rRNA, the UAAU-rich sequence (located about 10 bases upstream of the 5′ end of 28S-$\beta$ segment of 28S rRNA molecule) is where cleavage occurs by a 'late enzyme dependent cleavage event' to expose the loop (in stem and loop structure). Furthermore, in aphids the absence of the gap deletion can be explained by an absence of the UAAU-rich sequence necessary for gap processing (*Ogino et al., 1990*). From transcriptome analysis, we have obtained the 28S rRNA gene sequence for *Lepas anatifera* (Fig. 5A; currently in GenBank submission process). We have located the AU-rich region in the *Lepas* LSU, as well as the specific UAAU sequence which occurs upstream of a conserved motif 5′-CGAAAGGG-3′ (Fig. 5B) (*Fujiwara & Ishikawa, 1986*; *Van Keulen et al., 1991*; *Gillespie, Yoder & Wharton, 2005*). The UAAU stretch of rRNA starting at

**(A)**

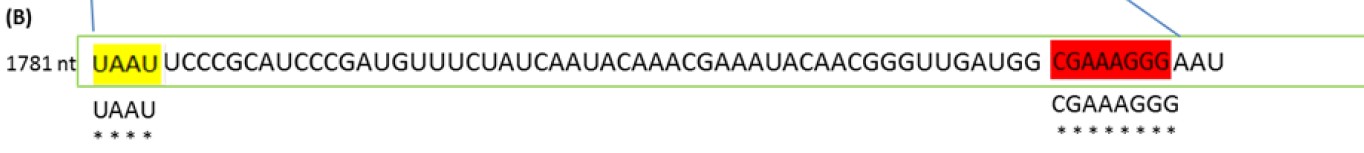

**(B)**

1781 nt

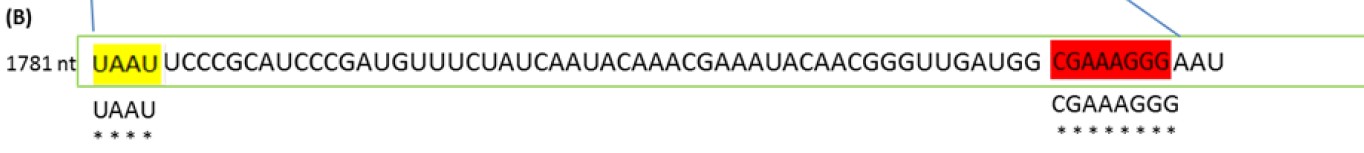

UAAU
\* \* \* \*

CGAAAGGG
\* \* \* \* \* \* \* \*

**(C)**

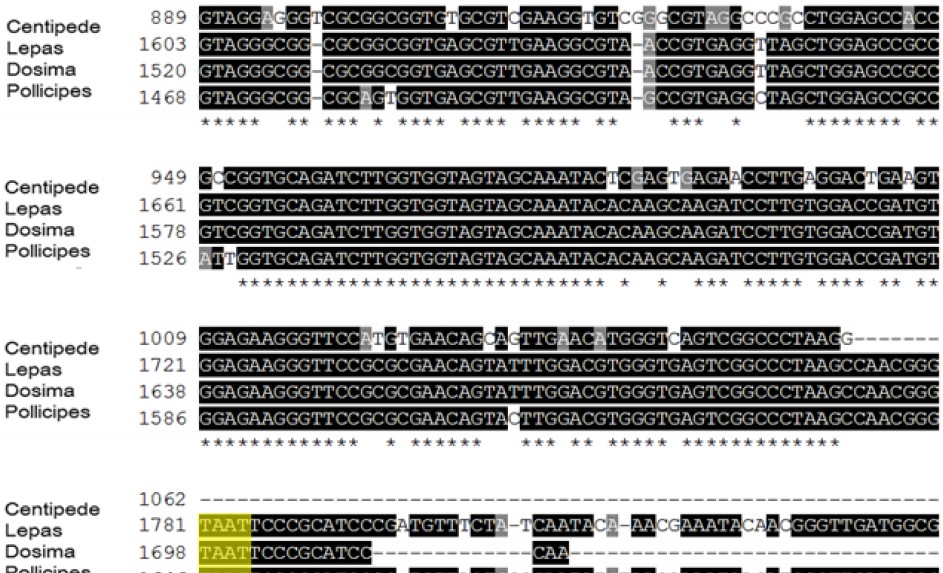

**Figure 5** **28S gene sequence analysis.** (A) 28S sequence for *Lepas anatifera* from Illumina Hi-Seq 2000 platform; (B) The AU rich stretch of rRNA where the UAAU sequence at position 1,781 nt is likely responsible for splitting the rRNA into ~1,780 nt and ~2,000 nt fragments which migrate together on a heat-denaturing gel. The CGAAAGGG sequence at the 3′ end of the AU rich region (highlighted in red) is highly conserved in all 28S rRNAs; (C) Sequence alignment for 28S rDNA of centipede and barnacles upstream of the TAAT (UAAU) cleavage site (yellow highlight). The 28S rDNA sequence for spider *Grammostola porteri* is not available.

position 1,781 nt in *Lepas* is likely responsible for splitting the rRNA into two fragments of ~1,780 nt and ~2,000 nt, which migrate together on a heat-denaturing gel due to their similar size. On the Agilent bioanalyser output (Fig. 2) we observed a double-headed peak for heat-denatured samples of the centipede and barnacles, which is likely due to the small difference in fragment size following cleavage. This UAAU sequence was present across the three barnacle species tested (Fig. 5C). While complete 28S rDNA across all four species with available sequence data was highly conserved (Fig. S1), the LSU sequence is not yet available for the *Grammostola* spider species.

The gap deletion is not thought to cause any functional problem for mature rRNA because unless denatured, the subunits remain bound together at the normal loop (*Sun et al., 2012*). But many researchers extracting RNA may experience practical concerns about apparent degradation: a Bioanalyser analysis of heat-denatured RNA is likely to yield an output reading such as 'RIN number not applicable (N/A)' due to complete absence of a detectable 28S fragment. In fact, these may be extractions of high quality and the sample just requires analysis in its non-heated form to yield a RIN. The present study adds to the knowledge about which organism groups display the 'degraded' RNA appearance on analysis. Researcher awareness of the 28s rRNA cleavage activated during the heat-denaturing process will eliminate unnecessary troubleshooting in the pursuit of high quality RNA profiles and RINs.

## ACKNOWLEDGEMENTS

We would like to thank our colleagues at School of Natural Sciences, NUIG and the barnacle stranding sightings community on 'theamazinggoosebarnacle' facebook page. Special thanks to the two reviewers whose comments were exceptionally helpful in improving the manuscript.

### Funding

This research was funded by Science Foundation Ireland, Contract grant number: 09RFPMTR2311 awarded to AMP http://www.sfi.ie/ and a Beaufort Marine Research Award grant-aided by the Marine Institute, Department of Communications, Marine and Natural Resources, Government of Ireland. The funders had no role in study design, data collection and analysis, decision to publish, or preparation of the manuscript.

### Grant Disclosures

The following grant information was disclosed by the authors:
Science Foundation Ireland: 09RFPMTR2311.
Marine Institute, Department of Communications, Marine and Natural Resources, Government of Ireland.

### Competing Interests

The authors declare there are no competing interests.

## Author Contributions

- Sean D. McCarthy conceived and designed the experiments, performed the experiments, analyzed the data, contributed reagents/materials/analysis tools, wrote the paper, prepared figures and/or tables, reviewed drafts of the paper.
- Michel M. Dugon contributed reagents/materials/analysis tools, wrote the paper, reviewed drafts of the paper, animal dissection and expertise.
- Anne Marie Power conceived and designed the experiments, analyzed the data, wrote the paper, prepared figures and/or tables, reviewed drafts of the paper.

## DNA Deposition

The following information was supplied regarding the deposition of DNA sequences:
GenBank, Lepas anatifera 28S sequence: KU052603.

## Supplemental Information

Supplemental information for this article can be found online at http://dx.doi.org/10.7717/peerj.1436#supplemental-information.

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
