# Peer review of "‘Degraded’ RNA profiles in Arthropoda and beyond"

_PeerJ, doi:10.7717/peerj.1436_

## Round 0.1 · original submission · Major Revisions

Please, consider all the suggestions in the revised versión of the manuscript.

Reviewer 1 ·

Basic reporting

Writing is clear enough. Figures are good. Article may have too little meat to be a unit of publication (see below).

Experimental design

Problem is described well enough and is important enough, technical work seems good as far as it went, but more experiments and more rigor would greatly improve the paper.

Validity of the findings

The conclusions are probably valid but not all alternate possibilities were ruled out, or all products identified. Thus, more experiments are suggested.

Additional comments

This paper seems to present the correct answer to a problem from past studies, and thus it is helpful. However, it is very “lightweight” and it could provide more conclusive results with a few more experiments and some more background research. Now it contains speculations and repetitions that will not be needed after the work is given added depth.

It is my style as a reviewer to write my general suggestions here in the commentary, and to pencil-mark my more detailed comments on the pages of a paper copy of the manusucript, to scan that and ask the editor to send it on to the authors.

More homework should be done on taxonomic naming and on the taxonomy of the arthropods. The traditional Linnean categories are no longer used much, and I found it jarring to have so many clades named ‘Subphylum this’ and ‘Class that.’ Maybe ‘Phylum’ is still acceptable, but can the authors check with a taxonomist on how to name the lower-level clades? For the arthropods specifically, the major clades currently recognized are Chelicerata, Myriapoda, and Pancrustacea (latter being insects within a paraphyletic crustacea). This is explained in the Regier et al. 2010 paper the authors cited, and even better is Rota-Stabelli et al. 2011 “A congruent solution to ...” in Phil. Trans. Roy. Soc. Lond. 278, 295.

I found the extra band at about 5000 or 6000 bases in the NorthernMax formaledehyde-denaturing gel to be very troubling (line 144, gel Figure 3, and perhaps in Figure 2). What is this band? Is it really DNA instead of RNA? Is the formaldehyde method invalid because it gives artifacts? I strongly suggest sequencing that band to find out what gene product it represents.

If the NorthernMax method gives artifacts, that also will be a significant finding of this study. Maybe it will better for investigators in the future to use heat-denaturing with the foreknowledge that three different rRNAs are in the so-called 18S peak, and then find some way to separate or sequence these rRNAs independently of one another?

Actually, I believe all the bands should be sequenced to confirm the interpretation. Confirm that they represent 18S, full 28S, 28S alpha and 28S beta, and rRNA versus DNA as claimed. Perhaps only about 200 or 300 nucleotides need to be sequenced per band to identify the products, which thus would not be too difficult. But universal confirmation by sequencing seems essential to me.

Apparently, the authors tested formaldehyde as the only non-heat denaturant, yet they draw conclusions about urea and other denaturants as well (around line 186). Can’t validly do that unless they also use urea, which I suggest they do, to expand the breadth of experiments. Also, use a series of urea and formaldehyde denaturants at different strengths, as the authors themselves suggest around line 190.

Around line 137, the authors use high RIN numbers of non-heat denatured samples to argue that heat denaturation does not degrade the rRNAs. That is a non sequitor; compares apples to oranges. I suspect the problem was that the measuring method cannot measure degradation, or calculate a RIN value, for heat-bisected rRNA - but it is still wrong to say non-heat RINs tell anything about heat degradation.

Can the authors be more precise about the location of the gap site in the 28S (LSU) rRNA? I think ~line 200 implies it is in divergent domain D7a (expansion segment 5?), in an AU-rich loop, but can that be stated more clearly, with the actual nucleotide-site numbers given, based on the honeybee sequence? Helpful references on this location are Gillespie J. et al. (2006) “Characteristics of the nuclear . . .” Insect Mol. Biol. 15, 657, and Mallatt J. and Chittenden, K. (2014). “The GC content of LSU rRNA evolves . . .” Mol. Phylogen. Evol. 72, 17. These papers could be cited.

The literature review and its table are okay, although maybe the language implies the deletion is more common among living organisms than it is? I made some suggestions on the pencilled pages about this.

In the last part of the paper, after line 221, could the authors explain more precisely how their recognition of the heat-denaturing problem can solve past problems with 28S rRNA?

Again, more experiments and more care and depth would greatly help this paper. I submit my detailed, pencil-marked comments with this review.

Annotated reviews are not available for download in order to protect the identity of reviewers who chose to remain anonymous.

·

Basic reporting

The content of the article is relevant to current RNA analysis and contributes to scientific knowledge. It is written in a simple language and its scope is appropriate for publication in peerJ journal.
Authors should however be careful with use of inappropriate words that may seem to rubbish other people's work e.g claims “Our findings are contrary to some claims in the literature (Krupp, 2005) ….”. The manuscript should be revised using appropriate language where applicable.

Experimental design

The design of the experiment should be improved by inclusion of more specimens.
Use of a single specimen for such analysis is not enough to make a solid conclusion. Biological replicates are required to rule out false positives. Thus the experiment should have been carried out with least duplicate samples.In case where the samples are limited, the authors should clearly explain in the manuscript

Validity of the findings

The findings of the study are sound. They could be supported with more data such as multiple sequence alignments of newly discovered gapped- rRNA with those of reported organisms. Sequences could be obtained from public biological databases such as GeneBank

Additional comments

N/A

---

## Round 0.2 · accepted · Accept

The Authors have improved the manuscript according to the reviewers comments.